# Lessons from an International Initiative to Set and Share Good Practice on Human Health in Environmental Impact Assessment

**DOI:** 10.3390/ijerph18041392

**Published:** 2021-02-03

**Authors:** Ben Cave, Ryngan Pyper, Birgitte Fischer-Bonde, Sarah Humboldt-Dachroeden, Piedad Martin-Olmedo

**Affiliations:** 1BCA Insight Ireland Ltd., D02FY24 Dublin, Ireland; ryngan.pyper@bcainsight.com (R.P.); birgittefischerbonde@gmail.com (B.F.-B.); 2International Association for Impact Assessment (IAIA), Fargo, ND 58103-3705, USA; sarahhumboldt92@gmail.com; 3European Public Health Association (EUPHA), Post Box 1568, 3500 BN Utrecht, The Netherlands; piedad.martin.easp@juntadeandalucia.es; 4Centre for Primary Health Care and Equity, University of New South Wales, Sydney, NSW 2052, Australia; 5Fischer-Bonde Consulting, 1727 Copenhagen, Denmark; 6Department of Social Science and Business, Roskilde University, 4000 Roskilde, Denmark; 7Escuela Andaluza de Salud Publica, 18011 Granada, Spain; 8Instituto de Investigacion Biosanitaria de Granada (Ibs. GRANADA), 18016 Granada, Spain

**Keywords:** Environmental Impact Assessment, human health, public health, Health Impact Assessment, Health in All Policies, significance, equity, health inequality, infrastructure

## Abstract

Environmental Impact Assessment (EIA) is applied to infrastructure and other large projects. The European Union EIA Directive (2011/92/EU as amended by 2014/52/EU) requires EIAs to consider the effects that a project might have on human health. The International Association for Impact Assessment and the European Public Health Association prepared a reference paper on public health in EIA to enable the health sector to contribute to this international requirement. We present lessons from this joint action. We review literature on policy analysis, impact assessment and Health Impact Assessment (HIA). We use findings from this review and from the consultation on the reference paper to consider how population and human health should be defined; how the health sector can participate in the EIA process; the relationship between EIA and HIA; what counts as evidence; when an effect should be considered ‘likely’ and ‘significant’; how changes in health should be reported; the risks from a business-as-usual coverage of human health in EIA; and finally competencies for conducting an assessment of human health. This article is relevant for health authorities seeking to ensure that infrastructure, and other aspects of development, are not deleterious to, but indeed improve, human health.

## 1. Introduction

We present lessons from drafting a technical reference paper [1] for health in Environmental Impact Assessment (EIA). From a core health sector perspective, this is quite a niche topic and yet EIA is a very important regulatory mechanism. In 2018, the UN Environment Programme [2] stated that EIA is the most commonly known, most commonly used, and globally widespread, environmental planning and management tool and that it is the only environmental policy tool that is required by most countries around the world and whose results are regularly publicly acknowledged and available. EIA is established in international and national law but it is applied at the local level and involves local government, civil society and the private sector. The theory and methods that inform EIA permeate thinking in spatial planning.

The broader context for this article is the way in which public health is considered in policy making, both in and outside of the health sector. This has direct relevance for Health in All Policies (HiAP) as it examines ways in which reasoning for public health is presented in a regulatory context outside the health sector. We paraphrase Morgan’s [3] wise observation that impact assessment practitioners and public health specialists have much to learn from one another and we look at topics such as the nature of evidence, the relative merits of quantitative versus qualitative methods, coping with uncertainty, the role, and effective use, of public participation, and evaluating social significance. This article has specific relevance for health authorities seeking to ensure that infrastructure, and other aspects of development, are not deleterious to, and indeed improve, public health.

### 1.1. Background

EIA is governed in the European Union (EU) by Directive 2011/92/EU on the assessment of the effects of certain public and private projects on the environment [4], as amended by 2014/52/EU [5] (hereafter the ‘EIA Directive’). The amendments made in 2014 include a stipulation that human health is considered when conducting EIA. Recital 41 of EU Directive 2014/52/EU states that the objective of EIA [is] to ensure a high level of protection of the environment and of human health.

In 2018, a joint action was initiated by the International Association for Impact Assessment (IAIA) and the European Public Health Association (EUPHA) to prepare a reference paper on health in EIA [1] and to set out ways in which the health sector can contribute to this regulatory requirement. A reference paper is a formal document that provides additional content which is relevant to existing commitments [6]. Reference papers tend to be non-binding but they can assist agencies in formulating approaches to a topic [7] and parties to an agreement can adopt them if they so choose [6]. They are often written before practices develop ‘by default’ into standards [8]. We present key findings from writing this reference paper. We start by introducing EIA and the amendments to the EIA Directive. We set out the materials and methods for this study and, in the discussion, we navigate the literature on policy analysis, Health Impact Assessment (HIA) and health in other forms of impact assessment. Whilst the reference paper is structured around the EIA Directive, the principles and approaches it discusses have global application to health in impact assessment.

### 1.2. Environmental Impact Assessment

EIA is a regulatory mechanism that was first introduced in the United States by the National Environmental Policy Act in 1969 [9] and it is now in place in many countries across the world. EIA is applied to projects such as transport infrastructure, residential development, infrastructure for processing waste materials and infrastructure for generating, transmitting and storing energy. These projects are typically large, with lasting effects on the environment. The European Commission states that the EIA Directive is a crucial tool for sustainable development [10]. The EIA Directive has currency outside the twenty-seven Member States because the financial institutions of the EU such as the European Bank for Reconstruction and Development (EBRD) and the European Investment Bank (EIB) ensure that projects to which they advance loans apply the EIA Directive when conducting environmental assessments [11,12].

The process of EIA is designed to sustain environmental values when developments that might compromise those values are proposed [13]. Box 1 shows the EIA process as set out in Article 1(2)(g) of the EIA Directive. (see Box 1).

There is a large but often well-defined cast of actors, or stakeholders, that take part in an EIA. The party that is seeking consent for a project will prepare an EIA Report. This party is typically referred to as the developer.

Box 1The EIA process as set out in Article 1(2)(g) of the EIA Directive. From Directive 2011/92/EU as amended by Directive 2014/52/EU [5].For the purposes of this Directive, the following definitions shall apply:[…]
(g)‘environmental impact assessment’ means a process consisting of:

(i)the preparation of an environmental impact assessment report by the developer, as referred to in Article 5(1) and (2);(ii)the carrying out of consultations as referred to in Article 6 and, where relevant, Article 7;(iii)the examination by the competent authority of the information presented in the environmental impact assessment report and any supplementary information provided, where necessary, by the developer in accordance with Article 5(3), and any relevant information received through the consultations under Articles 6 and 7;(iv)the reasoned conclusion by the competent authority on the significant effects of the project on the environment, taking into account the results of the examination referred to in point (iii) and, where appropriate, its own supplementary examination; and(v)the integration of the competent authority’s reasoned conclusion into any of the decisions referred to in Article 8a.


The EIA Report is examined by the competent authority which is the department of planning or the environmental regulator in a given geographic area. The competent authority then reaches a reasoned conclusion based on its examination of the developer’s EIA Report as well as the consultation responses, other documents submitted by the developer in support of the application and, where appropriate, the results of its own supplementary examination. Developers will often use private sector consultants to ensure they have expert advice on each of the topics listed in Article 3 (see Box 2).

The solutions proposed in the EIA tend to be avoidance or mitigation of adverse effects. Each assessment focusses upon effects that are both likely and significant (as per Article 5 (1)b). Attention is thus paid to identifying these potential adverse effects, and then finding ways in which to ensure that they do not occur. This is usually achieved by adapting the design of the project or reaching an agreement between the developer and the relevant authorities/stakeholders for measures to counteract, or mitigate, the adverse effects. These aspects are then monitored. EIA is a regulatory process so the changes to design and the agreed measures are set out in formal and binding agreements and will typically be funded by the developer.

### 1.3. Why Is the Amended EIA Directive Relevant to Human Health and to the Health Sector?

The 2014 amendments were transposed into the regulations of each EU Member State in 2017 [14]. The amendments are extensive, they affect most of the pre-existing legislation [15,16] and have considerable implications for EIA practice. Commentators note how the amendments require a focus on the quality of content in the EIA, on timeliness and relevance of reporting as well as monitoring project compliance [17]; how regards must be given to the potential effects of any different, or ‘alternative’, design scenarios that were considered [18] and how decision makers are required to engage with the public [19].

The main change, from a health perspective, is the inclusion of *human health* among the topics to be considered in EIA (see Box 2). It is this that brings the health sector into the orbit of EIA.

Box 2Text of Article 3 in the amended EIA Directive 2014/52 EU. From Directive 2011/92/EU as amended by Directive 2014/52/EU [5].
1.The environmental impact assessment shall identify, describe and assess in an appropriate manner, in the light of each individual case, the direct and indirect significant effects of a project on the following factors:
(a)population and human health;(b)biodiversity, with particular attention to species and habitats protected under Directive 92/43/EEC and Directive 2009/147/EC;(c)land, soil, water, air and climate;(d)material assets, cultural heritage and the landscape;(e)the interaction between the factors referred to in points (a) to (d).

2.The effects referred to in paragraph 1 on the factors set out therein shall include the expected effects deriving from the vulnerability of the project to risks of major accidents and/or disasters that are relevant to the project concerned.


Other topics are also introduced by the amendments: climate change mitigation and adaptation, accidents and disasters, land and biodiversity [18] (see Box 2). Hough [19] notes that expanding the factors to be considered will increase the workload of, and complexity in, preparing an EIA Report and that it can be expected to result in longer documents and a greater workload for impact assessors and public authorities. Mustow [20] notes that substantial changes in impact assessment practice are required to incorporate human health assessment. Change is not a problem in itself as EIA practice in Europe has been constantly developing since Directive 1985/337/EEC [21] made it mandatory for members of the European Economic Community. The 2014 amendments introduce human health as a new consideration and we look below at the implications of involving the health sector in the EIA process.

## 2. Materials and Methods

A writing team convened in September 2018. This team was consisted of members of the health section of IAIA and the HIA section of EUPHA. The World Health Organization (WHO) Regional Office for Europe provided support in the early stages. From the inception to the publication, in December 2020, there were three drafts of the reference paper and four consultation ‘events’ (see Figure 1) and the writing team met approximately 30 times.

The reference paper examines the EIA Directive and European Commission guidance on EIA [22,23,24] and regional and national guidance for health in EIA for example [25,26,27,28]. It complements guidance for HIA for example, [29,30] and it is informed by international good practice [31,32,33,34,35]. It builds on previous work conducted jointly, and separately, by IAIA, EUPHA and the WHO Regional Office for Europe [36,37,38]. It also draws on the literature on policy analysis, HIA and health in other forms of impact assessment and the experience of the writing team gained over two decades on work in public health and impact assessment.

The first draft of the reference paper was discussed in March 2019, at a technical meeting hosted by the European Centre for Environmental and Health at the WHO Regional Office for Europe in Bonn. Participants examined questions that had been identified through the practice of HIA, of health in EIA and the application of the amended EIA Directive. These are provided below and were first set out in Cave et al. [27] as questions to which the health sector in general, and public health in particular, needs to develop consensus to enable consideration of health in EIA.

How should population and human health be defined in EIA?How should the health sector participate in the EIA process?What is the relationship between EIA and HIA?What counts as evidence for changes in health?When is an effect ‘likely’ and ‘significant’?How should potential changes in health be reported?What are the risks from a business-as-usual coverage of population and human health in EIA?Who can conduct an assessment?

The subsequent consultation events invited comment on substantive issues to do with health in EIA and they also asked consultees about the clarity, structure, accessibility, etc. of the reference paper. The second draft was consulted upon, from November 2019 to January 2020, via a webinar and a survey hosted by IAIA. The third draft of the reference paper was peer reviewed, in March 2020, by three referees drawn from IAIA members before being finalized and then published in December of 2020. The draft reference paper was amended after each of these consultations.

A thematic coding strategy was applied to the comments provided in the consultation processes. This identified themes such as target audience of the reference paper; health determinants; significance/likely health effect; equity/vulnerable groups; methods/tools; role of public health professionals/competences needed; intersectoral cooperation/stakeholders/responsible authorities. The responses given to the survey and the peer-review comments were based upon a reading of the second and third drafts, respectively, of the reference paper and so these comments were also analyzed with the framework identified by Herber et al. [39] which relates to peer review of a qualitative manuscript. Details about the consultation and the responses are provided in the Appendix A.

## 3. Results

Figure 1 summarizes who took part in each round of consultation. We refer the reader to the Appendix A for details about the consultees as well as details about the ways in which the comments were taken into account. Table 1 summarizes the feedback given in the consultation. In drafting the reference paper, our focus was on presenting the role of the health sector in the amended EIA Directive while ensuring that the questions from Cave et al. [27] were addressed.

## 4. Discussion

The questions from Cave et al. [27] are discussed with reference to the consultation responses and the research literature below.

### 4.1. How Should Population and Human Health Be Defined in EIA?

The EIA Directive lists the topics that must be considered but it does not define these topics. So, this most basic of questions about defining population and human health sets parameters for the assessment as well as the stakeholders that take part in, and the expertise needed to conduct, the assessment. We look first at ‘population’ and then at ‘human health’.

In EIA ‘population’ effects are examined through socioeconomic analysis, and can include economic structure, labor markets, demography, housing, services (education, health, police, fire, etc.), lifestyles and values [40]. The juxtaposition of ‘population’ and ‘human health’ in Article 3 (1)a of the amended EIA Directive shows how ‘population’ and ‘human health’ are separate but linked. The health component of environmental assessment is usually applied at a population, and not an individual, level [41], p. 28. ‘Population health’ refers to the health outcomes of a group of individuals, including the distribution of such outcomes within the group [42].

How is health defined in impact assessment and in EIA? It depends who you ask. Those working in HIA and public health typically define health broadly, while a narrower definition has been found in EIA.

Key documents for HIA [31,32,33,35] cite the WHO constitution [43] which defines health as ‘a state of complete physical, mental and social well-being and not merely the absence of disease or infirmity’. This is supported by Health in All Policies (HiAP) and Healthy Public Policy (HPP) [44] and it traces a link between health outcomes and those factors in the social, physical and economic environment that influence, or determine, health outcomes.

In contrast, commentators in the following countries and jurisdictions report instances of a narrow interpretation of health within EIA, with a tendency to focus on biophysical aspects linked to the physical environment: Australia and New Zealand [3,45,46,47]; Canada [41]; Denmark [48]; France [49]; EU [50]; Italy [51,52]; Spain [53]; Sri Lanka and Wales [54]; Thailand [55]; and the US [56,57]. Seven of these articles focus on EIAs in European Member States [48,49,50,51,52,53,54], and five of these articles [48,49,52,53,54] are published after 2017, i.e., the date of transposition. The authors do not specify what Directive was being applied but it is likely that they are looking at EIAs that were carried out under the earlier Directive which had no explicit requirement to consider human health.

Critically, the different definitions of health and the divide between those working in public health and those working in EIA is not attributed to procedural compliance with the regulations and legislation which provide a mandate for EIA. It is instead attributed variously to the origins of the crossover between health and environmental assessment being in an environmental health tradition [58], to the predominance of the regulatory agencies in EIA, to the absence of health expertise in the EIA practitioner community and to entrenched EIA practices [56]. Commentators have noted how the absence of guidance documents addressing the particularities of health in EIA is a barrier to the inclusion of health into environmental assessments [48] and how this, in turn, leads practitioners to fall back on environmental health statutes to define the scope of the health concerns [56].

The definition in the WHO constitution balances the beneficial aspects of health and well-being with the need to address disease and infirmity. Consultees for this reference paper noted the importance of both parts of this definition and advised that ‘the absence of disease or infirmity’ emphasizes the importance of the healthcare sector, and ‘a state of physical, mental and social wellbeing’ emphasizes public health. Consultees also recommended including salutogenesis [59] and the cultural and spiritual dimensions to health. The decision was made to stay with the WHO definition as this encompasses these other concepts and it does not proscribe any topics from consideration in an assessment. We also placed value on the longevity of the WHO definition. As part of a text that was adopted in 1946, the WHO definition was in prominent use long before the drafting of the EIA Directive, and other environmental regulation citing both environment and health.

Thus, the scope of inquiry for health in EIA can be broad and include aspects such as healthy lifestyles, socioeconomic determinants, specialisms such as occupational health, as well as aspects resulting from exposure to physical contaminants and require input from toxicology, environmental epidemiology, etc. As we shall see below (see Section 4.5), consultees emphasized that starting with such a broad palette emphasizes the need to maintain a clear focus on the important health effects of a project.

Consultees noted the importance of ensuring the assessment examines inequalities in health and equity for vulnerable groups. We agree with Frohlich and Potvin [60] who state that a focus on vulnerable populations is complementary to a population approach and necessary for addressing social inequalities in health. Health equity introduces an explicit moral and ethical dimension to the way in which we look at health inequalities, i.e., whether the gap is unjust [61]. Health equity is both a principle and a goal. It motivates efforts to eliminate differences in health by improving the health of the economically/socially deprived. Participation and engagement, particularly for those with the least voice, lead to a more equitable process [62] and, in turn, to effective public policies [63]. An equitable process is a necessary precondition for equitable outcomes. Braveman [64] warns that public health policy can confuse equity with difference or inequality. Buse et al. [65] make a similar observation noting that HIA practice struggles with concepts such as fairness and justice.

We conclude by noting that, in 2001, European Commission guidance [66] to assess its own work programs required a broad definition of health and the consideration of inequalities. This guidance referred to the mandate, in the founding EU Treaty [67], for Union policies and activities to ensure a high level of human health protection.

### 4.2. How Should the Health Sector Participate in the EIA Process?

The WHO defines the health sector as consisting of organized public and private health services, health departments and ministries, health-related non-government organizations and community groups and professional associations adapted from source [68]. Given that the EIA Directive formally requires human health to be assessed, should the health sector participate and if so how and what parts of the health sector?

Should they participate? Consultees were clear that the health sector, from national to local, should take part in EIA and that this should be at the earliest stage possible. Banken [69] examines ways in which HIA can be mainstreamed, or institutionalized and refers to EIA as an example. He emphasizes the durability of legal frameworks for institutionalizing a permanent demand for a process such as HIA. He goes on to observe that even with legal obligations in place, ongoing input, support and quality control by public health would be needed to institutionalize HIA and to enable non-health actors to produce public health knowledge [69]. Simos et al. [70] also set out the importance of institutional support for health in impact assessment across the Healthy Cities European Network. Commentators frequently note the opportunities that HIA offers [36,71,72,73] as well as the way that HIA depends for its development upon the health sector at the regional and national [74] and global [75] levels. Taking a broader perspective, the case has been made that impact assessment has a capacity for transformative change in support of sustainable development [76]. Skills in impact assessment are transferable and can enhance what is currently seen as ‘core’ public health work. Authors have noted the contributions made by HIA to topics such as local planning [77,78,79]; to meeting the Sustainable Development Goals [80]; to the search for solutions to climate change [65]; and to addressing emerging infectious diseases [81] and across a wide range of sectors [82].

How should the health sector participate in EIA? Working across sectors is integral to an HiAP approach and should pose no conceptual problem to public health teams. However, the mandate to work in this way differs between countries and resourcing is an issue. de Leeuw [83] notes how cross sectoral efforts to address population health receive small amounts of funding in comparison to healthcare and how its advocates sit on the periphery of the policy playing field and political radar [83], a poetic turn of phrase that captures the liminal position of HIA [84]. Globally, public health teams working on impact assessment face resourcing challenges [48,56,70] in terms of finance and staff.

While the development and EIA processes are regulated by ministries and/or departments of environment or planning, health in EIA and HIA needs to be overseen by the health authority. Health authorities can provide input to EIA by reviewing the health component at various stages. They can also advise on sets of outcomes, pathways, effects on population health, follow up and mitigation and monitoring.

Roué Le Gall et al. [49] and Capolango et al. [85] describe the development of a guide for health and urban planning for use in both France and Italy and how the use of this guide is influenced by the ways in which health authorities can provide input to the EIA process in each Member State. In France, the environmental authority has to consult a health authority, and thus Roué Le Gall et al. [49] focused on the role played by the authorities reviewing the EIA Report rather than integrating health into the EIA Report. In Italy, the health authority’s statement is on a par with the environmental authority statement and both are sent to the developer [85].

Which parts of the health sector should contribute to EIA? Public health is typically the specialism to co-ordinate health input in EIA but all aspects of the health sector may be relevant from disease prevention, health protection and health promotion to health services. In the EU, the competence for public health rests with Member States and is articulated at regional and local levels and requires close working between the Union and the Member States. The work of public health is usually unseen but protecting health, promoting health and preventing and treating disease are core concerns for the health sector and to be truly effective, require a whole of government approach that spans different sectors.

The term ‘political will’ is often used to describe the support needed to justify time and resources for cross sectoral efforts such as HIA [53,74,75,86] and work towards healthy environments [87,88,89]. Commentators from a political science perspective question the analytical validity of ‘political will’ and instead draw attention to the ways that ‘interest groups’ present causes so they become politically acceptable [90]. It is also recognized that EIAs can be influenced by vested interests [46,91]. They are subject to political and economic forces, and are nested within complex political decisions and processes [92], and their content is tied to broader legislative requirements as well as policy and political decision making and additional approvals processes with stakeholders with conflicting agenda [93]. The process is often funded by the developer and so taking part in EIA can require the health authority to engage with the private sector. This factor alone causes nervousness for public health [53]. EIAs are typically carried out on contentious projects and the results of any assessment are likely to be contested.

Public health input is worthwhile to ensure mitigation is clearly understood, fit for purpose and able to be enforced. This also involves legal advice. Morris et al. [94] find that regulatory approaches to achieving a health dividend work in only a few situations: there needs to be an exposure that can be quantified and then a distinction made as to whether that exposure is acceptable in health terms; and there needs to be an adequately resourced enforcement agency, staffed by people with the knowledge and inclination to give priority to the issue in question. Where Morris et al. [94] focus on the health protection aspect, Polsky et al. [95] note the role of lawyers in facilitating a transition to HPP by identifying legal levers for changing business as usual. In the context of EIA, legal advice is relevant in drafting binding commitments at project level. For example, lawyers can advise developers on ways to engage in health-promoting activities without incurring undue legal liabilities and they can draft memoranda of understanding and/or funding agreements to institutionalize work across sectors. Legal advice is also relevant for setting the policy context in which a development takes place, for example Member States have the option to include the health authority in their national EIA legislation as formal consultees.

### 4.3. What Is the Relationship between EIA and HIA?

The relationship between EIA and HIA is of note for various reasons, not least because the health sector’s understanding of development and of impact assessment tends to be mediated through HIA. This topic is not explicitly addressed in the reference paper as consultees warned that explaining the differences between health in EIA and HIA was a distraction from the main objective of the reference paper namely to guide health authorities through the EIA process. Commentators [3,36], and consultees for the reference paper, note that methods and skills for HIA are a resource, and can support capacity building, for health in EIA.

HIAs reports may be commissioned alongside EIA Reports. HIA and EIA are distinct but similar processes. EIA is a statutory process with a defined procedure and output. Box 1 shows the EIA process as set out in Article 1(2)(g) of the EIA Directive. EIA is undertaken as a matter of law. EIA declares the likely significant effects of a project through an EIA Report. Health has to be integrated into EIA, but it can also be investigated in a separate HIA process that stands alone. There may be a policy requirement for a standalone HIA Report, or it may be undertaken as good practice to provide supporting analysis. Standalone HIA can have greater flexibility in its procedures and outputs. For example, standalone HIA typically takes a consensus-based approach and guidance documents refer to making *recommendations* which are typically non-binding.

Cole and Fielding [96] note that opportunities for HIA would be missed if it were only conducted when an EIA is conducted. Fehr et al. [36] caution against arbitrarily dividing health from other sectors and creating additional assessments and they note that the need and justification for standalone HIA cannot automatically be derived from the universally accepted significance of health. When an EIA follows good practice for human health, i.e., scoping the wider determinants of health and considering inequalities, then it is likely to fulfill HIA policy requirements. However, the EIA Directive (Annex IV) requires the likely significant health effects of the project to be reported within the EIA Report. The similarities between an EIA Report and a standalone HIA Report mean there are likely to be high degrees of duplication when both are undertaken in parallel. This will tend to increase costs and the burden on consultees and decision makers. However, the flexibility of standalone HIA to explore issues outside of the statutory remit of EIA should not be overlooked, e.g., strategic issues or recommendations based on community views beyond the issues scoped into an EIA.

McCallum et al. [97] set out three methodological considerations for enabling integration of health into EIA and smooth cross-sectoral work: (A) Ensure that the framework can be used as a stand-alone process and when integrated with EIA. Health in EIA needs to be consistent with HIA frameworks and to ensure ‘public health’ values [58] and approaches inform practice. The reference paper establishes overarching principles: a comprehensive approach to health; a focus on equity; an approach that is proportionate; and that is consistent with science while taking local context into account. (B) Apply language to closely align with EIA processes. This is the challenge of integration. How can the methodologies of a particular sector be fitted into the EIA Directive and national regulations? The reference paper examines the concepts used in, and the processes governing, EIA and considers how they can be interpreted and implemented for human health. (C) Devise a system for evaluating overall impact when a multitude of determinants are considered. A focus on one particular approach, for example quantitative analysis, can overlook this critical insight. Health in EIA requires a framework that applies to topics with different methodological and epistemological frameworks. A single assessment will present numerous findings on different topics and can therefore use both quantitative and qualitative approaches. EIA requires a clear and consistent focus on likely significant effects and reasoned conclusions on ways in which specific communities will be affected by specific project activities. These apply across a range of topics for example from air quality to employment or social cohesion. The reference paper sets out the conceptual models. This is covered in Section 4.5 below.

### 4.4. What Counts as Evidence for Changes in Health?

Impact assessment (IA) aims to reach objective, robust and evidence-based conclusions on the likely effects of a project [27]. The process and the outcomes of IA are thus concerned with scientific observation and analysis, with principles of design, with the application of regulations and law, and with the interpretation of local and contextual rights and understandings, so IA requires a broad range of activities that cuts across sectors and involves multiple stakeholders [76]. Sectors define evidence in different ways, for example, evidence for planners can refer to planning policy or to routine statistics, whereas for public health professionals evidence can also be retrieved from published academic literature [27]. There are different types of, and uses for, health evidence [98]. However, the evidence base relating to *specific* developments is often unlikely to be well developed or of the quality that is required to make definitive judgments [27].

Causal pathways are used in impact assessment and in risk assessment to trace the links between a cause and an effect. Causal pathways allow for complexity and for cumulative effects to be shown and to be examined [99], for uncertainty to be included at each stage of the assessment [100] and for community views and values to be included [101].

Analytical models used by public health to examine environmental effects on human health started with a ‘pollution-driven’ concept [102] and have expanded to include and identify opportunities for protection and prevention/improvement. The analytical models use quantitative and qualitative methods to establish the plausibility of a link between an event and an outcome. They include the Source–Pathway–Receptor model [103]; the Driver–Pressure–State–Impact–Response (DPSIR) model [104]; the Driver–Pressure–State–Exposure–Effect–Action (DPSEEA) model [105,106]; and subsequent developments of the DPSEEA model [94,102] which take on a socioecological model of public health. Findings from research into the exposome show promise. The exposome encompasses life-course environmental exposures, including lifestyle factors, from the prenatal period onwards [107] and requires involvement from a wide range of disciplines from exposure sciences, toxicology, biology, epidemiology, statistics, etc. [108], and latterly social [109] and economic [110] variables. This can show how environmental and social stressors converge in disadvantaged communities and it can provide decision makers with tools to measure these impacts [111].

In 1997, Davies and Sadler [41] stated that the assessment of health in environmental assessment needs significance criteria and indicators which cover a broad range of environmental hazards and which address social, community or psychological dimensions of health and well-being. For the psychological dimension, a survey may be required to understand community attitudes and risks to mental health [112].

It is clear that the techniques and methods for collecting information on health exist and so the salient step is to demonstrate the need for that information and how it is proportionate to the effects expected to arise from the project. This leads us onto *likelihood* and *significance*.

### 4.5. When Is an Effect ‘Likely’ and ‘Significant’?

The amended EIA Directive specifies that likely significant effects on the environment, including health, should be identified. This raises questions about both *likelihood* and about *significance*. We will also see that determining that an effect is *likely* and *significant* requires judgements to be made and thus we begin to see the role of values in these decisions, such as *importance, desirability* and *acceptability*.

Ehrlich and Ross [113] note that *likelihood* is a common element of significance determination in many jurisdictions, that this is a part of predicting the impact and should be done separately from determining the acceptability of the impact. They go on to note that for worst-case-type scenarios (meaning low-probability high-consequence events), even an unlikely impact may be unacceptable if it is severe enough. The degree of likelihood given to risks from zoonotic diseases and other emerging infectious diseases is one such, topical, example of an event that can be of low probability but have high consequences [114]. Ehrlich and Ross [113] state that *likelihood* should be understood in the context of risk when determining *significance*. Consultees noted that health pathway models can be used to establish *likelihood*. The reference paper incorporates risk analysis into a conceptual model for the scoping stage.

We consider *significance* before looking at ways in which findings on *significance* are presented. *Significance* is a central concept in impact assessment: it does not refer to statistical significance although that may be a way of demonstrating that a given effect is indeed significant. *Significance* draws attention to select issues so the competent authority can reach a reasoned conclusion as to whether that effect should be reduced or prevented [113].

Guidance which accompanies the amended EIA Directive brings the political dimension of significance into consideration and notes that a decision on *significance* relies on informed, expert judgement about what is *important*, *desirable* or *acceptable* with regards to changes triggered by the project in question [22,24]. Ehrlich and Ross [113] tell us that *significance* determinations are case-by-case applications of a value-based threshold which involves the comparison of a predicted change to a limit of acceptable change.

*Significance* is first considered, in a high-level manner, at the screening stage, e.g., the decision to state that further EIA work is needed requires a judgement that there are likely to be significant effects. *Significance* is examined in more detail at the scoping stage, where the assessor will provide a clear explanation of those effects that are considered potentially significant and which should therefore be ‘scoped in’ to the assessment, and those that are considered not significant and are thus ‘scoped out’ of the assessment. In the following impact assessment stage, each significant effect is looked at in detail and mitigation or enhancement measures are proposed to ensure that the effect is addressed appropriately. The *significance* of an effect therefore informs the design of the project and/or conditions that are needed to ensure that the project can go forward. This goes on to inform what is monitored.

There are many challenges to the ways that the *significance* of an effect is determined [115]. *Significance* may be interpreted differently across teams working on projects and also by external stakeholders [113,116]. When looking at how decisions on *significance* are reached, commentators have noted a failure to systematically address uncertainty [117,118,119]; a failure to recognize the political dimension of *significance* [120]; and a bias toward technical or quantitative analysis and positivistic reasoning at the expense of qualitative reasoning, contextual analysis, and the use of public knowledge and perspectives [56,119,121]. Singh et al. [122] report how both the reasoning process for determining *significance* and the level of input by stakeholders is opaque. Sadler [123] observed a tendency for assessors to shift the responsibility for making judgements on *significance* to decision makers.

Ehrlich and Ross [113] see that a strength of the determination of *significance* is that it is neither clear cut nor objective. It is a complex decision that is not based on a tick box approach or a simple application of standards and regulations. It is instead a professional judgement that is based on an informed and subjective judgement by decision makers. It uses cogent reasoning and it relies on evidence brought forth by the participants of the EIA. Importantly, this results in a decision that reflects the values of the person or organization making the decision and also ‘ideally … society’s values’ [113]. A determination of *significance* needs a nuanced understanding of the context in which decisions are made and the inclusion of society’s values as a defense of the democratic structure within which EIA sits [113]. It needs to be overarching, to acknowledge different inputs such as political context and local knowledge and it needs to be transparent.

In IA, the *significance* of an effect is typically established by identifying impact characteristics such as magnitude, duration, frequency, likelihood and reversibility and this is, or should be, applied uniformly across the topics in the EIA [113]. For health and well-being, *significance* is usually based on factors such as the magnitude or severity of the potential health effects; the number of people potentially affected; the size and nature of the potentially affected population(s), e.g., workers, children, the elderly, etc.; the frequency or duration of the potential health effects; the degree to which the health effects are reversible or irreversible; the probability or likelihood that the health effects will occur; and the level of uncertainty inherent in the health assessment [41], page 35. ICMM [124] addresses uncertainty by adding the degree of confidence in the impact occurring based on scientific and other evidence of the health impact occurring in similar circumstances elsewhere. The views of other stakeholders, including the public, need to be taken into account when determining *significance*. McCallum et al. [125] include public concern/interest as part of the scoping decision and Heller et al. [126] state that recommendations must focus on impacts to communities facing inequities and they must be responsive to community concerns. In addition to scientific literature, baseline conditions, consultation responses and regulatory standards, Pyper and Cave [127] include policy context and local and regional health priorities in a model for finding *significance*. This presents findings of *significance* within the EIA guidance framework [22,24] of *importance*, *desirability* and *acceptability*.

A finding of *significance* is a professional judgement which needs to be clear and transparent [41,124,125,127]. The reference paper provides a conceptual model which is based on Pyper and Cave [127] and is presented as one option for reaching a conclusion on *significance*. It can be applied to all determinants of health and allows for a clear presentation of the multiple criteria that can be applied when determining *significance*. It considers the magnitude of change and the sensitivity of the population; whether scientific literature shows a relevant causal relationship, or clear association, of sufficient effect size; the relevant health priorities in the study area; the health baseline and whether there is likely to be a substantial change, or even a small change in a large or highly vulnerable population; the relation of the project and its activities to existing health policy, particularly local health policies; themes in the consultation and whether there is consensus or disagreement; and regulatory thresholds or standards could be breached or nearly crossed (approached).

### 4.6. How Should Potential Changes in Health Be Reported?

The outputs of the assessment are presented in the EIA Report, as set out in Article 5 of the EIA Directive. The competent authority makes the EIA Report available for review and invites comment on the project and its environmental effects. The competent authority issues a reasoned conclusion on whether the project entails significant effects on the environment and, at risk of repetition, we remind the reader that the EIA Directive includes human health in its definition of the environment. This reasoned conclusion is based on an examination, which includes the EIA Report and the comments received during consultation. This reasoned conclusion must be incorporated into the final Development Consent decision. There are multiple stakeholders in EIA and therefore multiple audiences each with different expectations [76]. This has practical and methodological implications for the EIA as the developer and competent authority will want to see the process completed on time and for there to be a clear relationship between the topics raised by stakeholders and the EIA Report’s findings.

Consultees saw health in EIA as an opportunity to promote shared understanding between developers and health authorities. They noted that likely significant health effects, including cumulative effects, should be communicated to stakeholders, decision makers and the public. Review packages for environmental assessment [128] and for health in impact assessment [129,130] set out requirements for reporting including practical aspects such as a logical layout, a lay summary and ensuring that all sources are referenced and methodological aspects such as presenting information without bias and providing emphasis appropriate to its importance in the context of the EIA Report. Subjectivity is inherent to the EIA process [131] and consultees were clear that while it may not be possible to eliminate bias it is important to ensure that the process of the assessment should be transparent. Nieuwenhuijsen et al. [132] identified methodological needs for HIA as being novel participatory integrated full-chain HIA models, methods and tools that assess the full chain of events from initial planning decisions and scenarios linking sources, emissions, exposures and health impacts, and considering multiple exposures and complexities, interdependencies and uncertainties of the real world. We return to Bhatia and Wernham’s [56] observation of the importance of reasoning and of presenting a sustained argument for public health. Consultees noted the importance of presenting the analysis of multiple criteria clearly and how this can usefully be presented as a narrative and so the reference paper provides an example of a narrative.

In EIA topics, other than human health, the endpoints of analysis are typically changes in determinants of health, not changes in health outcomes. The assessment of human health requires a conclusion on how any change manifests itself in terms of health outcomes, e.g., a change in respiratory health, or mental well-being and this should include effects on inequalities in health and health equity. Ádám et al. [133] make the same point for the assessment of health in policies. The step from a health determinant to a health outcome raises the issue of attributing how the health outcomes in a future population will change as a result of the proposed development that is undergoing an EIA.

Health is a cross-cutting issue and so its relation with other topics in the assessment needs to be considered. Health assessment often takes the outputs of assessments of, for example, air quality or socioeconomic change, as its starting point. This can reduce the time available for the health assessment. It can place the health assessment in a receptive mode that is dependent on decisions taken in other topics. As the assessment of human health becomes more embedded into EIA then the exchange between topics should become more equal with requirements of human health, e.g., a focus on inequality and equity, influencing the methodology of other topics. There can also be tension when there is a difference between emerging scientific research and guidance from bodies such as the World Health Organization and limit values set by EU directives.

### 4.7. What Are the Risks from a Business-As-Usual Coverage of Population and Human Health in EIA?

As discussed above (see Section 4.1), the way in which health is addressed in EIA is not due to procedural compliance with the regulations and legislation which provide a mandate for EIA but to tradition [58], to the regulatory agencies in EIA, to the absence of health expertise in the EIA practitioner community and to entrenched EIA practices [56]. This suggests that in the absence of guidance, the business-as-usual coverage of population and human health in EIA risks defaulting to low levels of input from the health sector and variable methods and standards in the way that health is addressed and judged. Wood et al. [134] looked at early experience across the EIA sector in the UK from July 1988, when EIA became mandatory, to December 1990. They characterized the response as inexperienced developers commissioning inexperienced consultants to prepare EIA Reports for submission to inexperienced competent authorities. To some extent, this is where we are currently with human health in EIA. Consultees noted that there may be a failure to identify the health implications of a project, that opportunities to protect and improve population health may be missed and that EIA compliance issues may be raised if not all the likely significant health effects of a project are reported within the EIA Report. Morgan [3] warns that stand alone, project-level HIA, with minimal links to formal impact assessment processes and decision making, runs the risk of increasing the marginalization of health issues, and even lowering the standing of health in the eyes of decision makers through ineffective practices. There are lessons to be learnt from the inclusion of human health in the Strategic Environmental Assessment (SEA) Directive [135] in 2004. While it did lead to health appearing in SEA reports [136], it did not lead to a growth in participation by the health sector [137]. As shown earlier, leaving human health to environment specialists leads to a narrow interpretation of human health.

That said, impact assessment commentators have called for formal health input to EIA, as have HIA and public health commentators. It is also important to recognize the work of the environment sector and the challenge for IA professionals to take on health as a topic with minimal input and support from the health sector. There is the potential for the IA workforce to add their knowledge and expertise to that of the public health workforce. Guidance exists in different regions and Member States and work is needed to track the ways in which these are affecting the coverage of health in EIA.

### 4.8. Who Can Conduct an Assessment?

How are standards set for the EIA process and how are they interpreted for health? EIA relies upon high-quality EIA Reports that can contribute to sound decision making. The content in the EIA Report must be prepared by ‘competent experts’ and the competent authority’s review (examination) requires ‘sufficient expertise’. Writing about HIA, Birley and colleagues [138,139] also note the need for a division of competencies between the managers and the consultants who deliver HIA and the government officers who regulate HIA. EIA health competencies have yet to be formally defined. Looking at health and environmental assessment in Spain, Iglesias-Merchan et al. [53] note that capacity development is not cheap and that health is playing catch-up with environmental assessment which is established both in university courses and as a profession. We also add that there is a considerable body of research looking critically at EIA and a growing body of research looking at health in impact assessment.

It is good practice for those involved in health in EIA, on behalf of the developer and on behalf of the competent authority, to have knowledge of impact assessment, public health and environmental sectors. Balance is needed in setting the bar appropriately given the need for capacity.

Public health competencies comprise soft skills, such as leadership, advocacy and intersectoral consensus building, and technical skills, ranging from epidemiology and natural sciences to ethics and sociology. Consultees noted that technical skills in impact assessment include assessment of specific determinants as well as knowledge of the EIA process, and the legal and ethical dimensions. For health in EIA, a public health background is desirable with knowledge and skills across relevant health determinants. A team should have mixed skills and the ability to translate and adapt to the technical demands for different sectors that bring forward EIA projects.

Fothergill and Marshall [140] report on schemes for certifying the skills in EIA and they also note anecdotal evidence that organizations across Europe are investigating the potential to develop professional EIA recognition schemes as a result of the amendments to the EIA Directive. Consultees noted how competence frameworks for public health [141,142,143,144] could be merged with competence frameworks for impact assessment.

## 5. Conclusions

We present lessons from drafting a technical reference paper [1] for health in EIA. EIA is not perfect but it is an established and practical process for working through the dialogue needed for sustainability. Arabadjieva [15] observes that the amendments to the EIA Directive encourage deeper engagement with environmental issues by developers and competent authorities. The EIA Directive specifies that health is part of the environment and that human health is one of the topics to be considered. We make the case for adopting a broad definition of health in EIA and for addressing health inequalities and health equity.

Impact assessment practitioners cannot do this alone. The best-placed specialism to coordinate health input to EIA is public health, and this, in turn, needs institutional support from the wider health sector at the regional, national and global levels. The impact assessment workforce could be called on to add their knowledge and expertise to that of the public health workforce. We note the wider benefits to health sector involvement in EIA: impact assessment has a transformative capacity to build knowledge about sustainability; and EIA is a mechanism by which the whole of society responds to existing local priorities and to issues such as climate change and emerging infectious diseases. Public health input to EIA will help meet the Directive’s objective of a high level of health protection and it will contribute to a more complete enactment of the EIA Directive.

The reference paper explains the process of EIA and an overarching methodology for all determinants of health. There remains a need for guides that support analytical methods for individual determinants of health, e.g., socioeconomic, air quality or transport. Case studies of health in EIA were requested by consultees but, in this mostly unfunded endeavor, a search for and presentation of case studies was beyond the capacity of the writing team. Future case studies could include examples from specific sectors and regions.

In looking at ways in which a reasoned public health argument can be advanced and in showing the ways in which significance and uncertainty can be addressed in any one context, this article offers a contribution to the ways in which society can tackle policy challenges across health and the environment. EIA involves drafting binding agreements and so, after Coggon et al. [145], we note that it provides examples of ways in which legal and other regulatory measures can both serve and restrict the pursuit of agenda in public health.

The understanding of factors that influence human health continues to deepen and to jump ahead of policy. Initiatives such as the Commission on the Social Determinants of Health [146], concepts such as One Health [147] and Planetary Health [148], and emerging scientific research into political [149] and commercial determinants of health [150] as well as the exposome [109] show how analysis of health requires an understanding of the links, and differences, between individuals, communities and their wider physical, social and economic environment at all scales from global to molecular. EIA provides a structured and proportionate way to examine a particular project in a particular context while taking account of a wide range of information including science and local knowledge.

## Figures and Tables

**Figure 1 ijerph-18-01392-f001:**
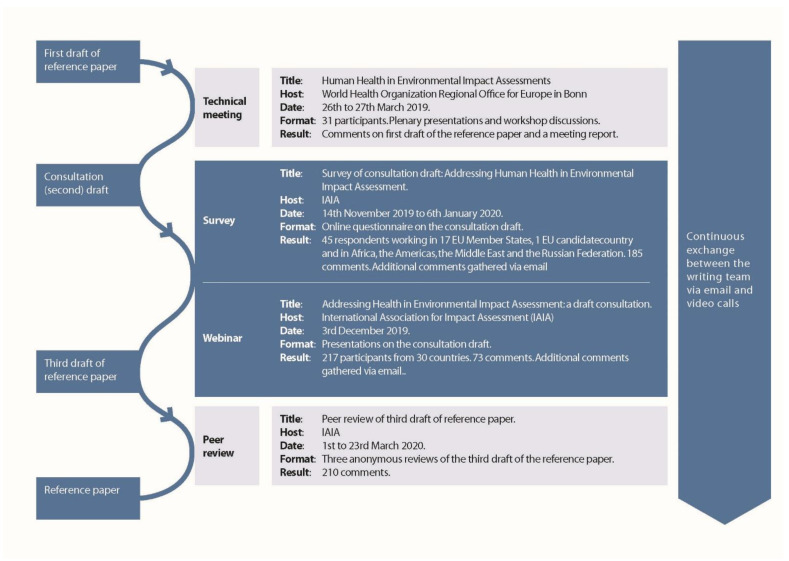
Preparing the reference paper. Figure from Cave et al. [1]. Reproduced with permission from Cave et al. Human Health: Ensuring a High Level of Protection. A Reference Paper on Addressing Human Health in Environmental Impact Assessment as Per EU Directive 2011/92/EU Amended by 2014/52/EU; published by International Association for Impact Assessment and European Public Health Association, 2020.

**Table 1 ijerph-18-01392-t001:** Summary of themes from the consultation.

Dimensions	Themes
How should population and human health be defined in EIA?	The EIA definition of environment includes human health. Use the WHO definition of health, i.e., ‘physical, social, mental health and wellbeing’ and ‘absence of disease and infirmity’. Use the wider determinants of health, i.e., environmental, social and economic. Discuss risk factors and public understanding of risk. Take a public health approach. Consider inequalities in health and equity for vulnerable groups. Be consistent in the approach to health in EIA.
How should the health sector participate in the EIA process?	Public health should be engaged throughout EIA. Health experts should be particularly involved at the EIA scoping stage. Be proportionate in scoping health determinants. The health authority can provide advice and identify data sources. Health authorities should review EIA Report health chapters. Resources are needed for health input to EIA.
What is the relationship between EIA and HIA?	EIA is governed by EU legislation whereas standalone HIA is not. Methods and skills for HIA are a resource, and can support capacity building, for health in EIA. Both HIA and EIA rely on intersectoral working and cooperation. There is potential for duplication if an HIA is conducted at the same time as an assessment of health within an EIA.
What counts as evidence for changes in health?	Use best available evidence and peer-reviewed scientific literature. Qualitative and quantitative approaches are both valid. Evidence informing significance includes scientific literature, health priorities, baseline health data, policy review, consultation results and regulatory standards.
When is an effect ‘likely’ and ‘significant’?	‘Likelihood’ is separate from ‘significance’. Health pathway models can be used to establish likelihood. ‘Significance’ in EIA is distinct from ‘statistical significance’. Significance for health in EIA … requires a model/framework that spans all determinants of health; considers whether health change is important, desirable or acceptable; is based on professional judgement and evidence; depends on context; and takes account of the sensitivity of a population and the magnitude of impact.
How should potential changes in health be reported in EIA?	Presenting the assessment of human health in the EIA Report is clearer if there is a dedicated chapter for human health. Focus on the ‘likely’ and ‘significant’ health effects of a project. Examine the effects on inequalities in health. Differentiate between ‘risk factors’ and ‘determinants of health’. The process of assessment should be transparent. Consider opportunities for promoting/improving population health as well as protection. Consider cumulative effects to population health from multiple stressors. Reporting should be evidenced based and it should enable reasoned conclusions to be made. Report health ‘effects’ in terms of ‘health outcomes’. Present the analysis of multiple criteria as a narrative. Communicate the health effects to stakeholders, decision makers and the public. Promote shared understanding between developers and health authorities. Use monitoring proportionately to track significant adverse health effects. Provide clarity on how to monitor, including governance for managing effects. Acknowledge that some settings have limited resources and poor data quality.
What are the risks from a business-as-usual coverage of population and human health in EIA?	A body of practice may emerge where the health sector is not engaged in EIA. There may be a failure to identify the health implications of a project. Opportunities to protect and improve population health may be missed. EIA compliance issues may be raised if not all the likely significant health effects of a project are reported within the EIA Report.
Who can conduct an assessment?	Health in EIA requires expertise in public health and impact assessment. Make use of the, currently separate, competence frameworks for public health and for impact assessment. Technical skills include assessment of specific determinants as well as knowledge of the EIA process, legal and ethical dimensions. Soft skills include health advocacy and intersectoral consensus building. Multi-disciplinary teams provide a breadth of skills for EIAs across different sectors.

## Data Availability

Not applicable.

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
