# Peer review of "Lessons from an International Initiative to Set and Share Good Practice on Human Health in Environmental Impact Assessment"

_ijerph, 2021, doi:10.3390/ijerph18041392_

Round 1

Reviewer 1 Report

I am very supportive of this paper. It represents a very strong contribution to the field of HIA. My only request is the author's consider recent guidance which has been developed and not considered as part of this work accomplished by the Asian Development Bank. While the commissioner of the HIA Sourcebook is Asia focused it does contain an important chapter on core competencies for HIA that could strengthen this work.

Excellent work. 

Reviewer 2 Report

The item of the paper is very interesting and the examination is rigorously conducted.

I suggest some changes to gain impact, to make the text more effective

  • The title: Ensuring a high level of protection of public human health. Lessons from preparing a reference paper on human health in environmental impact assessment
  • keywords: Environmental Impact Assessment; Public human Health; HIA  Health Impact Assessment; Significance
  • Abstract:The European Union Directive 2011/92/EU as amended by 2014/52/EU requires likely significant effects of a project on human health to be considered in Environmental Impact Assessments (EIA). Recital 41 of EU Directive 2014/52/EU states that the objective of EIA [is] to ensure a high level of protection of the environment and of human health.In 2018 a joint action was initiated by the International Association for Impact Assessment (IAA) and the European Public Health Association (EPHA) to prepare a reference paper on health in EIA. This aimed to establish basic definitions and common principles for public health in EIA, to set out ways in which the health sector can contribute to this regulatory requirement and to complement existing guidance. We navigate literature on policy analysis, Health Impact Assessment (HIA) and health in other forms of impact assessment. We consider ways in which society can solve policy challenges across health and the environment, and ways in which legal and regulatory measures can both serve and restrict the pursuit of a public health agenda...show how analysis of public health requires an understanding of the links, and differences, between individuals, communities and their wider physical, social and economic environment at all scales from global to molecular...The article is relevant for public health authorities seeking to ensure that infrastructure, and other aspects of development, are not deleterious to, but indeed improve, human health.

Moreover, keeping in mind that:  …”In looking at ways in which a reasoned public health argument can be advanced and in showing the multiple criteria that can inform the case for public health in any one context, this article has a broader contribution to the ways in which society can solve policy challenges across health and the environment, and after Coggon et al [124], to ways in which legal and other regulatory measures can both serve and restrict the pursuit of agendas in public health”….

I suggest you check every time you use the word health and replace it with public health.

Reviewer 3 Report

1. Please rephrase abstract. Reduce introductory content and propose importance.

2. In the keywords, HIA should not be written as abbreviation.

3. The definition of health in 4.1 should be enhanced.

4. In part 4.4, the ‘changes’ could be explained in detail.

5. In part 4.7, ‘In the absence of guidance, the business-as-usual coverage of population and human health in EIA risks defaulting to low levels of input from the health sector and variable methods and standards in the way that health is addressed and judged’, could you please analyze the reason.

6. As a review paper, this title should be rectified to highlight it’s features clearly.

Reviewer 4 Report

Overall, it’s great that you documented this process! In general, help your audience understand how to use these findings. How can an EIA/ HIA practitioner or scholar easily track on your lessons and implement/advance them? They may disregard them as related to EU or WHO initiatives but they seem relevant for specific nations or communities, too. I would try to speak to this broader audience and hold our hands a bit more.

Some suggestions to this end:

  • I would reconsider the title. For those that don’t know what a reference paper is, it doesn’t capture the significance of your work. It sounds like you are reflecting on writing just any old paper or lit review but you are actually reflecting on a rigorous process designed to inform policy and international conversations. I would pick a title that more so reflects this for a global audience.
  • Reread for awkward phrasing and spelling (e.g., focusses) and clarify some language throughout. For instance, you say that EIA “was first brought in.” What do you mean by this? What do you mean by “competent authorities”?
  • Can you say more about your methods? You say, “The technical meeting sought guidance on the direction that the reference paper should take. Participants examined questions that had been identified through the practice of HIA, of health in EIA and the application of the amended EIA Directive.” Can you say more about who and how? Also, say a bit more about the “thematic coding strategy [that] was applied to the comments provided in the consultation processes.”
  • As you discuss many frameworks/initiatives (e.g., Commission on the Social Determinants of Health [125]; concepts such as One Health [126] and Planetary Health [127]; emerging scientific research into political [128] and commercial determinants of health [129] as well as the exposome [94]…) you may want to also discuss EIA and HIA as tools for addressing cumulative impacts. In the U.S. at least, risk assessment that informs decision-making is inherently flawed as it does not prioritize health inequities. EIA and HIA could, and this may be worth acknowledging more so. (See Solomon & Morello-Frosch)
  • You may want to refer to Frohlich & Potvin, 2008 when discussing this perpetual issue: “7. What are the risks from a business-as-usual coverage of population and human health in EIA?” Often well-intentioned interventions can perpetuate inequity.
  • In your conclusion, you say, “There is a need for the WHO to match its public statements and assume its leadership role in supporting health in impact assessment.” Can you add a bit more to reemphasize your main suggestions? “There is a need for the WHO to match its public statements and assume its leadership role in supporting health in impact assessment by doing X, Y, and Z….”

Again, this paper seems worth sharing but help readers connect with it more so. 

Round 2

Reviewer 4 Report

The authors have effectively addressed reviewer comments.